# The Risk of Venous Thromboembolism after Thoracolumbar Spine Surgery: A Population-Based Cohort Study

**DOI:** 10.3390/jcm12020613

**Published:** 2023-01-12

**Authors:** Hao-Wen Chen, Wen-Tien Wu, Jen-Hung Wang, Cheng-Li Lin, Chung-Yi Hsu, Kuang-Ting Yeh

**Affiliations:** 1Institute of Medical Sciences, Tzu Chi University, Hualien 970, Taiwan; 2Department of Orthopedics, Hualien Tzu Chi Hospital, Buddhist Tzu Chi Medical Foundation, Hualien 970, Taiwan; 3School of Medicine, Tzu Chi University, Hualien 970, Taiwan; 4Department of Medical Research, Hualien Tzu Chi Hospital, Buddhist Tzu Chi Medical Foundation, Hualien 970, Taiwan; 5Management Office for Health Data, China Medical University Hospital, Taichung 404, Taiwan; 6Graduate Institute of Clinical Medical Science, China Medical University, Taichung 404, Taiwan; 7Graduate Institute of Clinical Pharmacy, Tzu Chi University, Hualien 970, Taiwan

**Keywords:** venous thromboembolism, deep vein thrombosis, pulmonary embolism, thoracolumbar spine surgery

## Abstract

Background: Although venous thromboembolism (VTE) is rare, including deep vein thrombosis (DVT) and pulmonary embolism (PE), it is a catastrophic complication after spinal surgery. This study was aimed to investigate the risk factors and incidence of VTE after thoracolumbar spine surgery (TLSS). Methods: We retrieved the data of 8697 patients >20 years old who underwent TLSS between 2000 and 2013 from Taiwan’s Longitudinal Health Insurance Database 2000. Each patient was randomly frequency-matched with four individuals who did not undergo TLSS by age, sex, and index year (the control group). Results: The incidence rates of VTE in the TLSS and control groups were 1.84 and 0.69 per 1000 person-years, respectively. The TLSS group had a higher VTE risk (adjusted HR (aHR): 2.13, 95% confidence interval [95%CI]: 1.41–3.21), DVT (aHR: 2.20, 95%CI: 1.40–3.46), and PE (aHR: 1.60, 95%CI: 0.68–3.78) than the control group. The correlated risk factors of VTE included older age (50–64 years: aHR: 2.16, 95%CI: 1.14–4.09; ≥65 years: aHR: 3.18, 95%CI: 1.65–6.13), a history of cancer (aHR: 2.96, 95%CI: 1.58–5.54), heart failure (aHR: 2.19, 95%CI: 1.27–3.78), and chronic kidney disease (aHR: 1.83, 95%CI: 1.18–2.83). Conclusions: The overall VTE risk following TLSS was less than 2% but correlated with certain risk factors. This information could help the spine surgeon help the patient prevent this fatal complication.

## 1. Introduction

With advancements in surgical techniques and devices, an increasing number of degenerative spine surgeries have been performed over the past two decades in developed countries, including the United States [1,2,3]. Postoperative venous thromboembolism (VTE), including deep vein thrombosis (DVT) and pulmonary embolism (PE), is a rare but lethal complication and can lead to a high mortality rate after thoracolumbar spine surgery (TLSS) [4,5,6]. However, VTE incidence has been reported to vary considerably in different populations and countries [7,8]. Wołkowski et al. in 2020 claimed that the sum score of the patient and surgical procedure characteristics was important in the evaluation of the risk of VTE [9]. In this period of the COVID-19 pandemic, Mureșan et al. in 2022 concluded that nutrition status, peripheral arterial disease, malignancy, and tobacco may predict the of risk and mortality of PE and DVT among COVID-19 patients [10]. Although the pharmacological prophylaxis suggested by the American College of Chest Physicians and the European Society of Anaesthesiology has generally become a routine protocol, another serious complication, epidural hematoma, may still occur (incidence rate: 0.3%) and induce permanent neurological injury when this protocol is initiated [11,12]. Knowing the risk factors related to the perioperative VTE is very important for orthopedic surgeons, especially when they plan to perform the elective operations for the patients, such as total joint arthroplasty or thoracolumbar spine surgeries, etc. Kawai et al. in 2020 and Melinte et al. in 2022 revealed that older age, malignancy, cardiovascular disease, obesity, tobacco, steroid, and low preoperative function score may be the important risk factors for the incidence of DVT during total hip or knee arthroplasty [13,14]. Identifying populations with a higher risk of postoperative VTE is critical for the patients who are scheduled to receive thoracolumbar spine surgery but only a few population-based studies have evaluated the incidence and risk of postoperative VTE in TLSS [15,16]. Therefore, in this study, we estimated the nationwide incidence of VTE in patients who underwent TLSS in Taiwan and attempted to identify the risk factors affecting VTE incidence.

## 2. Materials and Methods

For this study, after obtaining approval from the Institutional Review Board of China Medical University (CMUH104-REC2-115-CR2), we collected data from the Longitudinal Health Insurance Database 2000, which includes the detailed health-care data from 1996 to 2011 of 1 million beneficiaries who were randomly selected from the 2000 Registry for Beneficiaries of Taiwan’s National Health Insurance Research Database in 2000. In order to protect the privacy of the patients, all the personal information has been encoded for the research. The diagnostic codes used in this study were based on the International Classification of Diseases, Ninth Revision, Clinical Modification (ICD-9-CM).

In this study, we compared the VTE risk between individuals who underwent elective TLSS (the TLSS group) and those who did not (the non-TLSS control group). For the TLSS cohort, we enrolled adults aged >20 years who underwent TLSS, including corrective osteotomy for one vertebral segment (surgical code: 64269B), anterior spinal fusion (surgical code: 83043B), anterior spinal fusion for more than six levels (surgical code: 83044B), posterior spinal fusion (surgical code: 83045B), posterior spinal fusion for more than six levels (surgical code: 83046B), and laminectomy for decompression (surgical code: 83002C) under a diagnosis of thoracolumbar spondylosis (ICD-9-CM codes: 721.3, 721.42, 722.10, 722.51, 722.52, 722.73, 722.93, 724.02, 724.2, and 724.3) whose data were added to the database from 2000 to 2012. Postoperative day 1 after TLSS was set as the index date. We excluded patients with a history of spinal trauma (ICD-9-CM: 805, 806) and hospital admission for VTE, DVT (ICD-9-CM: 453.8), or PE (ICD-9-CM: 415.1) within a year before inclusion. We also excluded patients with a history of any spinal procedure, arthroplasty surgery, or fixation surgery of the lower limbs. We then selected the non-TLSS control group from the same database by age and gender matching with the TLSS group initially to the eighth digit and then, as required, to the first digit. The comorbidities of the patients included in this study included cancer (ICD-9-CM: 140–208), heart failure (ICD-9-CM: 428), atrial fibrillation (ICD-9-CM: 427.31), coronary artery disease (ICD-9-CM: 410–414), hypertension (ICD-9-CM: 401–405), chronic kidney disease (ICD-9-CM: 585), cerebral vascular accident (ICD-9-CM: 430–438), deep venous thrombosis (ICD-9-CM: 453.8), pulmonary embolism (ICD-9-CM: 415.1), chronic obstructive pulmonary disease (ICD-9-CM: 490–492, 494, and 496), diabetes mellitus (ICD-9-CM: 250), paralysis (ICD-9-CM: 342–344), and lower leg trauma (ICD-9-CM: 820, 821, 823, ICD-9-CM procedure codes: 81.51–81.54). The follow-up was stopped if patients suffered from DVT or PE, which we defined as the primary outcomes of this study.

### Statistical Analyses

Continuous variables were summarized as means and standard deviations, and categorical variables were listed as number of cases and percent values. Continuous between-group variables were compared using a Student’s *t*-test, and categorical variables were assessed using either a chi-squared test or a Fisher’s exact test. The variables in the multivariable model included age, sex, comorbidities of cancer, heart failure, atrial fibrillation, coronary artery disease, hypertension, chronic kidney disease, cerebral vascular accident, chronic obstructive pulmonary disease, diabetes mellitus, paralysis, and lower leg trauma or surgery, which were significantly different in the univariate Cox’s model. All statistical analyses were performed using SAS v9.4 (SAS Institute, Cary, NC, USA). The significance level was set at 0.05, and all tests were two-tailed.

## 3. Results

We included 8697 patients in the TLSS group and 8697 individuals in the age- and gender-matched non-TLSS group (Table 1). The mean age was 58.5 ± 14.1 years in the TLSS cohort and 57.8 ± 14.5 years in the non-TLSS cohort. In the TLSS cohort, 26.7% of the patients were aged 20–49 years, and 47.9% were men. The comorbidity without significant differences between the TLSS group and control group was cancer (3.43% vs. 3.73%). The comorbidities with significant differences between the TLSS group and control group included heart failure (6.22% vs. 2.87%), atrial fibrillation (1.39% vs. 0.93%), coronary artery disease (30.3% vs. 11.2%), hypertension (53.5% vs. 30.7%), chronic kidney disease (13.5% vs. 4.98%), cerebral vascular accident (6.36% vs. 4.84%), chronic obstructive pulmonary disease (19.1% vs. 7.54%), diabetes mellitus (14.5% vs. 7.03%), paralysis (4.95% vs. 1.55%), and lower leg trauma or surgery (2.53% vs. 0.93%) (Table 1). The mean follow-up duration until DVT and PE occurrence was 5.93 (SD = 3.59) and 5.96 (SD = 3.59) years, respectively, in the TLSS cohort and 5.79 (SD = 3.53) and 5.80 (SD = 3.53) years, respectively, in the non-TLSS cohort.

Overall, the incidental rates of VTE were 1.84 and 0.96 per 1000 person-years in the TLSS and non-TLSS cohorts, respectively (Table 2). After adjustment for age, sex, and comorbidities, the VTE risk was significantly higher in the TLSS cohort than in the non-TLSS cohort (aHR: 2.13, 95%CI: 1.41–3.21, *p* < 0.001). The relative VTE risk in the age-specific adjustment was significantly higher in patients aged 50−64 years (aHR: 2.16, 95%CI: 1.14–4.09, *p* < 0.05) and ≥65 years (aHR: 3.18, 95%CI: 1.65–6.13, *p* < 0.001) than in those aged ≤49 years (Table 2). Moreover, the relative risk of developing VTE in the comorbidity-specific adjustment was higher in patients with cancer (aHR: 2.96, 95%CI: 1.58–5.54, *p* < 0.001), heart failure (aHR: 2.19, 95%CI: 1.27–3.78, *p* < 0.01), and chronic kidney disease (aHR: 1.83, 95%CI: 1.18–2.83, *p* < 0.001; Table 2).

The overall incidences of DVT of the TLSS cohort and the control cohort were 1.45 and 0.58 per 1000 person-years, respectively. After adjustment for age, sex, and comorbidities, the risk of DVT remained significantly increased in the TLSS cohort compared with the control cohort (aHR: 2.20, 95%CI: 1.40–3.46, *p* < 0.001) (Table 3). When the sex-specific TLSS cohort was compared with the non-TLSS cohort, the relative risk of DVT was significant for men (aHR: 3.61, 95%CI: 1.63–8.00, *p* < 0.01). The relative risk of DVT in the age-specific TLSS compared with the control cohort was significantly higher for the aged ≥65 groups (aHR: 2.07, 95%CI: 1.12–3.83, *p* < 0.05). DVT risk stratified by having any comorbidity was higher in patients with TLSS than in patients without TLSS (aHR: 2.00, 95%CI: 1.20–3.33, *p* < 0.01). The cumulative incidence of DVT was significantly higher in the TLSS cohort than in the control cohort (*p* < 0.001; Figure 1a).

The overall incidence of PE was higher in the TLSS cohort than in the control cohort (0.46 and 0.16 per 1000 person-years, respectively) without statistical significance (Table 3). The risk of PE in the TLSS cohort was also not significantly different in the age, gender, and comorbidity subgroups of the two cohorts. However, the cumulative incidence of PE was still significantly higher in the TLSS cohort than in the control cohort (*p* = 0.006; Figure 1b).

## 4. Discussion

The results of this nationwide retrospective cohort revealed that the real-world incidence of VTE after TLSS is approximately 0.184% person-years. Most studies have evaluated thromboembolic risk only after specific spinal surgeries in a smaller case series, resulting in a wide range of variations in the observed incidence rates. According to a recent meta-analysis by Zhang et al., the overall incidence of VTE is 0.35%; the analysis revealed wide variation (0.15–29.38%) in the findings of the original studies [17]. This may be because different screening methods were used, as was reported in a systematic review by Glotzbecker et al., who further noted that the incidence of postoperative VTE (mainly DVT) was much higher when venography (12.3%) and ultrasonography (3.74%) were used than when clinical screening was used alone. The higher incidence of venography or ultrasonography may also be the result of inclusion of more asymptomatic cases or patients with minor symptoms, who require only observation and supportive care [8].

The relatively lower VTE incidence rate of the present study may also be explained by several other factors, including the limited availability of epidemiological data in Asia, underdiagnoses, low awareness of the thrombotic disease, and potentially less symptomatic VTE in Asian patients [18,19,20,21]. Almost all patients in Taiwan’s NHIRD are of an East Asian ethnicity. Differences in predisposition to VTE have been reported to exist in various Asian subpopulations, including ethnic Chinese populations [22,23,24]. The theory of the genetic risk of thrombophilia in Asian populations indicated by Wang et al. may explain the lower incidence rate in this study. In Asian populations, several genetic abnormalities, including mutations in factor V and prothrombin and deficiencies in protein S, protein C, and antithrombin, weaken the coagulation pathway. Moreover, factor V Leiden and prothrombin G20210A polymorphisms were found to be more exclusive to White than to Asian people, whereas the prevalence of protein S, protein C, and antithrombin deficiencies were found to be higher in Asian populations than in White populations [23,24,25,26,27,28,29]. However, a meta-analysis of 12 studies explored the factors predicting VTE after surgery for degenerative spinal diseases and reported that the VTE incidence in Asian patients (7.5%) was significantly higher than that in White patients (1%) [30], which is inconsistent with previous studies. Thus, ethnic differences in VTE incidence remain controversial. Although the inherited risk factors for VTE are different between Asian and White populations and should be explored in larger studies, the major acquired risk factors are similar in both populations [31].

According to Fuji et al., a history of VTE leads to a risk of recurrent VTE [32]. To differentiate TLSS-associated VTE risk from VTE recurrence, we excluded patients with a history of VTE. Patients with heart failure or chronic kidney diseases who have undergone TLSS are likely to have limited to no mobility, poor venous return, and fluid accumulation, each of which can independently induce venous stasis and VTE. This may be compounded by inherent TLSS-related reasons for an increased risk of postoperative VTE: (a) prolonged immobilization of the limbs during operation; (b) decreased postoperative mobilization, similar to that of other arthroplasty surgeries; and (c) direct manipulation of the large vessel when the retroperitoneal approach is employed [33,34,35]. In the present study, three of the patient-related risk factors were associated with an increased VTE risk after thoracolumbar surgery: history of cancer, heart failure, and chronic kidney disease. Chronic heart failure with falling LVEF can decrease patients’ exercise tolerance and mobility [36]. The fluid accumulation and poor venous return occurring in patients with chronic kidney disease or renal failure can similarly increase the VTE risk [37,38].

A study revealed a nonsignificant lower risk (odds ratio: 0.76, 95%CI: 0.18–3.18) of VTE in hospitalized medical patients with renal failure. However, considering the significantly higher bleeding tendency in hospitalized patients with renal failure, which is mainly the result of impaired renal function (total odds ratio: 1.43, 95%CI: 1.06–1.93), most chemical prophylaxis, such as low-molecular-weight heparins, would be a contraindication, leaving this population in high danger of thromboembolic-related risk [39,40]. In addition, compared with patients with normal kidney function, those with chronic kidney disease (CKD) were reported to have a higher VTE risk and to be more vulnerable to CKD progression. In a study on patients with VTE, moderate to severe CKD was associated with an increased death risk, VTE recurrence rate, and significant bleeding rate compared with mild to no CKD [40].

Our findings revealed that patients with a history of cancer had a higher risk of VTE, likely resulting from the inherent pathogenetic mechanisms in primary cancers. Moreover, cancer treatments, such as chemotherapy, radiotherapy, and target therapy, cause vascular toxicity and hypercoagulability, resulting in a substantially increased risk of VTE [41]. Studies have reported that active cancer and a static history of cancer have different levels of VTE risk. A 2020 systematic review and meta-analysis revealed that compared with individuals with no history of cancer, patients with active cancer, defined as the presence of cancer on admission or within the past year, had a lower risk of VTE (OR: 2.65, 95%CI: 1.79–3.91) than those with a nonactive history of cancer (OR: 3.20, 95%CI: 2.14–4.79), although both exerted a moderate risk [38]. Our study revealed that the overall adjusted risk of a history of cancer was approximately three times (aHR: 2.96) higher in the TLSS than in the non-TLSS group. This was likely because we considered a history of cancer to be a confirmed pathological diagnosis in a past medical record and, therefore, to include both ongoing and nonactive cancers, which is likely why the magnitude of the risk of our study cohort was between the aforementioned reported values for patients with cancer and those with nonactive cancer. A different incidence rate was also obtained in another study using a large health-care claims database; the study analyzed the data of patients in the United States who were commercially insured between 2004 and 2009 and reported an overall VTE development rate over 12 months after chemotherapy initiation in patients with solid cancers (bladder, colorectal, lung, ovary, pancreatic, or gastric cancers) that was approximately nine times higher (12.6% vs. 1.4%, *p* < 0.001) than previously reported for patients without cancer [42]. Thus, not only cancer status, including ongoing or nonactive, but also cancer histology or type and treatment can interfere with VTE risk estimation. Further research on stratified cancer status and histological diagnosis is warranted to identify patients with specific cancer types or receiving particular cancer-related therapy who might benefit from more aggressive prophylaxis.

In addition to the risk factors for VTE identified in our study, other risk factors were identified in a retrospective cohort study based on a nationwide readmission database. In the study, despite the relatively low readmission rate with a VTE diagnosis within 30 days (0.42%) and 90 days (0.62%) in patients undergoing degenerative spine surgery, patients undergoing thoracolumbar surgery of advanced age, with prolonged length of stay, using corticosteroids, and with a disposition to institutional care were deemed to be at a higher risk of complicated VTE and fatal outcomes [43]. VTE incidence assessed using color Doppler ultrasonography was reported to not be related to the severity of paralysis but rather primarily to age, particularly among populations with an acute thoracolumbar spinal cord injury, which is partially consistent with the finding regarding the age-related adjusted VTE risk (50−64 years, aSHR: 2.16, 95%CI: 1.14–4.09; ≥65 years, aSHR: 3.18, 95%CI: 1.65–6.13) of the present study [44].

This study has several limitations. First, information about the diagnostic methods used to identify the thromboembolism was not available for every case on the NHIRD. This study used the ICD-9-CM codes registered on the NHIRD, and patients who qualified as having a diagnosis of VTE were required to have received at least two outpatient treatments or one inpatient treatment for VTE. Although the National Health Insurance Administration regularly reviews and audits the consistency of the coding with the therapy regimen and the laboratory data of each individual, potential coding errors might still be present. Second, we used the ICD-9-CM codes to identify the comorbidities of the included patients. The detailed medical conditions may not be considered thoroughly. For example, we did not divide cerebral vascular accident into cerebral hemorrhage and cerebral infarction, which may have different effects on the incidence of VTE. Third, although a history of cancer was identified as a major risk factor, we did not perform stratification analyses by cancer status, location, or therapy. Nevertheless, the overall VTE risk was significantly higher in the patients with a history of cancer, which is consistent with the findings reported in previous studies. Future studies should perform subgroup analyses of cancer status (e.g., based on active or nonactive status or cancer stage), location (such as in the lung, breast, or prostate), and therapy (such as radiotherapy, chemotherapy, or target therapy). Finally, the included patients underwent anterior, posterior, or combined fusion of the thoracolumbar spine. Because data on the extent of spine correction and the instruments used were unavailable in the NHIRD, we could not analyze certain surgical factors, such as the operation time, intraoperative blood loss, and postoperative immobilization period, that may affect VTE incidence. Future studies should stratify patients undergoing different kinds of TLSS.

## 5. Conclusions

The incidence density rates of VTE were 1.84 and 0.96 per 1000 person-years for the TLSS and non-TLSS cohorts, respectively. The relative adjusted VTE risk was significantly higher in older patients (age > 50 years). Patients with an underlying comorbidity had a higher risk of post-TLSS VTE. In particular, a history of cancer, heart failure, or chronic kidney disease was an independent risk factor for VTE. Because prophylaxis is often administered to patients during the perioperative period, future studies should investigate in detail the pros and cons of various mechanical or chemoprophylactic treatments to identify and better treat populations vulnerable to serious complications after TLSS.

## Figures and Tables

**Figure 1 jcm-12-00613-f001:**
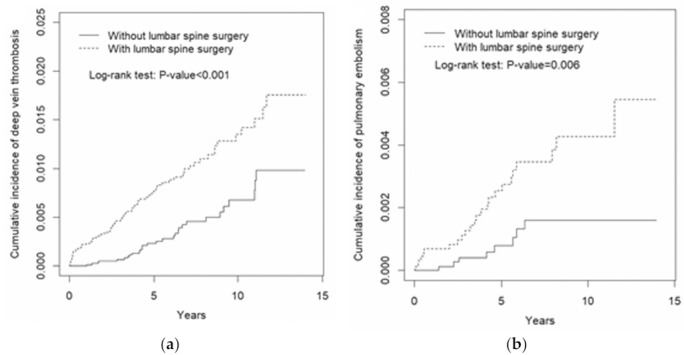
Cumulative incidence of (**a**) deep vein thrombosis and (**b**) pulmonary embolism in patients with thoracolumbar spine surgery and those without thoracolumbar lumbar spine surgery.

**Table 1 jcm-12-00613-t001:** Comparison of demographics and comorbidity between thoracolumbar spine surgery patients and controls.

Variable	Thoracolumbar Spine Surgery		*p*-Value
	No	Yes	
	(*n* = 8697)	(*n* = 8697)	
	*n* (%)	*n* (%)	
Age, years			0.99
≤49	2325 (26.7)	2325 (26.7)	
50–64	3252 (37.4)	3252 (37.4)	
≥65	3120 (35.9)	3120 (35.9)	
Gender			0.99
Female	4532 (52.1)	4532 (52.1)	
Male	4165 (47.9)	4165 (47.9)	
Comorbidity			
Cancer	324 (3.73)	298 (3.43)	0.29
Heart failure	250 (2.87)	541 (6.22)	<0.001
Atrial fibrillation	81 (0.93)	121 (1.39)	0.01
Coronary artery disease	975 (11.2)	2631 (30.3)	<0.001
Hypertension	2669 (30.7)	4654 (53.5)	<0.001
Chronic kidney disease	433 (4.98)	1178 (13.5)	<0.001
Cerebral vascular accident	421 (4.84)	553 (6.36)	<0.001
Chronic obstructive pulmonary disease	656 (7.54)	1661 (19.1)	<0.001
Diabetes mellitus	611 (7.03)	1264 (14.5)	<0.001
Paralysis	135 (1.55)	433 (4.98)	<0.001
Lower leg trauma or surgery	81 (0.93)	220 (2.53)	<0.001

Chi-square test; *t*-test.

**Table 2 jcm-12-00613-t002:** Incidence and Hazard ratio for VTE and VTE-associated risk factor.

Variable	Event	PY	Rate ^#^	Crude HR (95%CI)	Adjusted HR ^§^ (95%CI)
Thoracolumbar spine surgery					
No	35	50,395	0.69	1.00	1.00
Yes	95	51,559	1.84	2.65 (1.80, 3.91) ***	2.13 (1.41, 3.21) ***
Age, year					
≤49	13	30,811	0.42	1.00	1.00
50–64	46	38,833	1.18	2.82 (1.52, 5.22) **	2.16 (1.14, 4.09) *
65+	71	32,309	2.20	5.28 (2.92, 9.55) ***	3.18 (1.65, 6.13) ***
Sex					
Female	83	53,487	1.55	1.60 (1.12, 2.29) **	1.33 (0.92, 1.91)
Male	47	48,466	0.97	1.00	1.00
Comorbidity					
Cancer					
No	119	99,482	1.20	1.00	1.00
Yes	11	2471	4.45	3.74 (2.01, 6.96) ***	2.96 (1.58, 5.54) ***
Heart failure					
No	111	98,236	1.13	1.00	1.00
Yes	19	3718	5.11	4.55 (2.79, 7.41) ***	2.19 (1.27, 3.78) **
Atrial fibrillation					
No	126	101,103	1.25	1.00	1.00
Yes	4	850	4.71	3.78 (1.40, 10.2) **	1.52 (0.54, 4.30)
Coronary artery disease					
No	80	82,179	0.97	1.00	1.00
Yes	50	19,775	2.53	2.60 (1.83, 3.71) ***	1.06 (0.70, 1.62)
Hypertension					
No	45	61,797	0.73	1.00	1.00
Yes	85	40,156	2.12	2.92 (2.03, 4.19) ***	1.29 (0.83, 2.00)
Chronic kidney disease					
No	99	93,498	1.06	1.00	1.00
Yes	31	8455	3.67	3.47 (2.32, 5.20) ***	1.83 (1.18, 2.83) **
Cerebral vascular accident					
No	119	97,320	1.22	1.00	1.00
Yes	11	4633	2.37	1.94 (1.05, 3.61) *	0.97 (0.51, 1.84)
Chronic obstructive pulmonary disease					
No	104	89,429	1.16	1.00	1.00
Yes	26	12,524	2.08	1.78 (1.16, 2.74) **	0.86 (0.54, 1.35)
Diabetes mellitus					
No	106	92,106	1.15	1.00	1.00
Yes	24	9847	2.44	2.13 (1.36, 3.31) ***	1.14 (0.72, 1.82)
Paralysis					
No	126	99,040	1.27	1.00	1.00
Yes	4	2913	1.37	1.08 (0.40, 2.92)	
Lower leg trauma or surgery					
No	125	100,337	1.25	1.00	1.00
Yes	5	1616	3.09	2.49 (1.02, 6.08) *	1.36 (0.55, 3.37)

* *p* < 0.05, ** *p* < 0.01, *** *p* < 0.001. VTE: venous thromboembolism. Rate ^#^: incidence rate, per 1000 person-years; Crude HR: represented hazard ratio; Adjusted HR ^§^: hazard ratio adjusted for age, gender, comorbidities of cancer, heart failure, atrial fibrillation, coronary artery disease, hypertension, chronic kidney disease, cerebral vascular accident, chronic obstructive pulmonary disease diabetes mellitus, paralysis, and lower leg fracture or surgery.

**Table 3 jcm-12-00613-t003:** Incidence and hazard ratio for DVT and PE stratified by age, gender, and comorbidity for patients with thoracolumbar spine surgery compared to those without thoracolumbar spine surgery.

Variable	Thoracolumbar Spine Surgery						Crude HR (95%CI)	Adjusted HR ^§^ (95%CI)
	No			Yes				
	(*n* = 345,793)			(*n* = 345,793)				
	Event no	PY	Rate	Event no	PY	Rate		
DVT	29	50,401	0.58	75	51,607	1.45	2.52 (1.64, 3.87) ***	2.20 (1.40, 3.46) ***
Age, year								
20–49	3	15,354	0.20	9	15,464	0.58	2.98 (0.81, 11.0)	2.83 (0.73, 10.9)
50–64	9	19,410	0.46	27	19,442	1.39	3.00 (1.41, 6.37) **	2.09 (0.94, 4.66)
65+	17	15,637	1.09	39	16,701	2.34	2.13 (1.20, 3.76) **	2.07 (1.12, 3.83) *
Gender								
Women	21	26,284	0.80	43	27,238	1.58	1.98 (1.17, 3.33) *	1.65 (0.94, 2.91)
Men	8	24,117	0.33	32	24,369	1.31	3.93 (1.81, 8.54) ***	3.61 (1.63, 8.00) **
Comorbidity ^§^								
None	10	30,905	0.32	10	17,739	0.56	1.69 (0.70, 4.06)	2.17 (0.89, 5.29)
With any one	19	19,496	0.97	65	33,868	1.92	1.97 (1.18, 3.28) **	2.00 (1.20, 3.33) **
PE	8	50,444	0.16	24	51,820	0.46	2.93 (1.32, 6.52) **	1.60 (0.68, 3.78)
Age, year								
20–49	1	15,360	0.07	2	15,471	0.13	1.99 (0.18, 22.0)	1.36 (0.11, 17.0)
50–64	2	19,419	0.10	10	19,517	0.51	4.98 (1.09, 22.7) *	2.70 (0.56, 13.0)
65+	5	15,665	0.32	12	16,831	0.71	2.22 (0.78, 6.30)	1.31 (0.41, 4.21)
Gender								
Women	8	26,309	0.30	16	27,365	0.58	1.93 (0.83, 4.51)	0.82 (0.32, 2.10)
Men	0	24,135	0.00	8	24,454	0.33	-	-
Comorbidity ^§^								
None	3	30,927	0.10	2	17,761	0.11	1.18 (0.20, 7.04)	1.60 (0.26, 9.84)
With any one	5	19,517	0.26	22	34,059	0.65	2.52 (0.96, 6.66)	2.56 (0.97, 6.77)

* *p* < 0.05, ** *p* < 0.01, *** *p* < 0.001. PY: person-years; Rate: per 1000 person-years; Crude HR represented hazard ratio; Adjusted HR ^§^: hazard ratio adjusted for age, gender, comorbidities of cancer, heart failure, atrial fibrillation, coronary artery disease, hypertension, chronic kidney disease, cerebral vascular accident, chronic obstructive pulmonary disease, diabetes mellitus, paralysis, and lower leg fracture or surgery; DVT: deep vein thrombosis; PE: pulmonary embolism.

## Data Availability

The data set used in this study is managed by Taiwan’s Ministry of Health and Welfare (MOHW). Researchers can request access to this data set by application to the MOHW. Please contact the MOHW (email: stcarolwu@mohw.gov.tw) for further details (address: No. 488, Sec 6, Zhongxiao E. Rd, Nangang Dist, Taipei City 115, Taiwan, ROC). Phone: þ886-2-8590-6848). All relevant data analyzed in this article are presented herein.

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
