# Peer review of "The Risk of Venous Thromboembolism after Thoracolumbar Spine Surgery: A Population-Based Cohort Study"

_jcm, 2023, doi:10.3390/jcm12020613_

Round 1
Reviewer 1 Report
It is very important to conduct such a complete enumeration. As mentioned in limitation, it is difficult to ascertain the real incidence. However, it is reasonable to consider risk factors for incidence.
The content is generally fine, but there are a few things need to be added.
1. What is a "Cerebral vascular accident"? The pathophysiology is very different between cerebral infarction and cerebral hemorrhage.
2. There are some significant differences factors were not mentioned. Even if the report is similar to other reports, please add it in a simple manner.
Author Response
It is very important to conduct such a complete enumeration. As mentioned in limitation, it is difficult to ascertain the real incidence. However, it is reasonable to consider risk factors for incidence.
The content is generally fine, but there are a few things need to be added.
- What is a "Cerebral vascular accident"? The pathophysiology is very different between cerebral infarction and cerebral hemorrhage.
Ans: Thank you for your reminding. We used ICD-9-CM codes: 430-438; ICD-10 codes: I60-I69, G45, G46 as the presence of cerebral vascular accident as past history of the patients. We did not differentiate this code into cerebral infarction and cerebral hemorrhage. Since this was not statistically significant in the adjusted analysis, we will add this into our limitation section as below” Second, we used the ICD-9-CM codes to identify the comorbidities of the included patients. The detailed medical conditions may not be considered thoroughly. For example, we did not divide cerebral vascular accident into cerebral hemorrhage and cerebral infarction, which may have different effects on the incidence of VTE.”
- There are some significant differences factors were not mentioned. Even if the report is similar to other reports, please add it in a simple manner.
Ans: Thank you for your reminding. We have revised and supplemented our results as below: “We included 8697 patients in the TLSS group and 8697 individuals in the age-and -gender–matched non-TLSS group (Table 1). The mean age was 58.5 ± 14.1 years in the TLSS cohort and 57.8 ± 14.5 years in the non-TLSS cohort. In the TLSS cohort, 26.7% of the patients were aged 20–49 years, and 47.9% were men. The comorbidity without significant differences between TLSS group and control group was cancer (3.43% vs. 3.73%). The comorbidities with significant differences between TLSS group and control group included heart failure (6.22% vs. 2.87%), atrial fibrillation (1.39% vs. 0.93%), coronary artery dis-ease (30.3% vs. 11.2%), hypertension (53.5% vs. 30.7%), chronic kidney disease (13.5% vs. 4.98%), cerebral vascular accident (6.36 % vs.4.84%), chronic obstructive pulmonary dis-ease (19.1% vs. 7.54%), diabetes mellitus (14.5% vs.7.03%), paralysis (4.95% vs.1.55%), and lower leg trauma or surgery (2.53% vs 0.93%) (Table 1). The mean follow-up duration until DVT and PE occurrence was 5.93 (SD = 3.59) and 5.96 (SD = 3.59) years, respectively, in the TLSS cohort and 5.79 (SD = 3.53) and 5.80 (SD = 3.53) years, respectively, in the non-TLSS cohort.
Overall, the incidental rates of VTE were 1.84 and 0.96 per 1,000 person-years in the TLSS and non-TLSS cohorts, respectively (Table 2). After adjustment for age, sex, and comorbidities, the VTE risk was significantly higher in the TLSS cohort than in the non-TLSS cohort (aHR: 2.13, 95%CI: 1.41–3.21, p<0.001). The relative VTE risk in the age-specific adjustment was significantly higher in patients aged 50−64 years (aHR: 2.16, 95%CI: 1.14–4.09, p<0.05) and ≥ 65 years (aHR: 3.18, 95%CI: 1.65–6.13, p<0.001) than in those aged ≤ 49 years (Table 2). Moreover, the relative risk of developing VTE in the comorbidity-specific adjustment was higher in patients with cancer (aHR: 2.96, 95%CI: 1.58–5.54, p<0.001), heart failure (aHR: 2.19, 95%CI: 1.27–3.78, p<0.01), and chronic kidney disease (aHR: 1.83, 95%CI: 1.18–2.83, p<0.001; Table 2).
The overall incidences of DVT of the TLSS cohort and the control cohort were 1.45 and 0.58 per 1,000 person-years, respectively. After adjustment for age, sex, and comorbidities, the risk of DVT remained significantly increased in the TLSS cohort compared with the control cohort (aHR: 2.20, 95%CI: 1.40–3.46, p<0.001) (Table 3). When the sex-specific TLSS cohort was compared with the non-TLSS cohort, the relative risk of DVT was significant for men (aHR:3.61, 95%CI: 1.63–8.00, p<0.01). The relative risk of DVT in the age-specific TLSS compared with the control cohort was significantly higher for the aged ≥ 65 groups (aHR: 2.07, 95% CI: 1.12–3.83, p<0.05). DVT risk stratified by having any comorbidity was higher in patients with TLSS than in patients without TLSS (aHR: 2.00, 95%CI: 1.20–3.33, p<0.01). The cumulative incidence of DVT was significantly higher in the TLSS cohort than in the control cohort (p < 0.001; Figure 1a).
The overall incidence of PE was higher in the TLSS cohort than in the control cohort (0.46 and 0.16 per 1,000 person-years, respectively) without statistical significance (Table 3). The risk of PE in the TLSS cohort was also not significantly different in the age, gender, and comorbidity subgroups of the two cohorts. But the cumulative incidence of PE was still significantly higher in the TLSS cohort than in the control cohort (p = 0.006; Figure 1b).”

Reviewer 2 Report
This is an interesting and well written paper that fits the Journal scope with a very large population enrolled in the study. Methods are adequate and well explained. Results are clearly presented and well discussed. Referencing to previous literature is adequate as well. I have no hesitation to recommend straightforward publication.
I have only few suggestions. The authors can improve the Introduction section with recent paper published in literature regarding the risk of VTE, as following:
- https://doi.org/10.3390/medicina58101502
- https://doi.org/10.3390/jcm9123970
- https://doi.org/10.3390/diagnostics12112757
- https://doi.org/10.3390/jcm9051257
Author Response
This is an interesting and well written paper that fits the Journal scope with a very large population enrolled in the study. Methods are adequate and well explained. Results are clearly presented and well discussed. Referencing to previous literature is adequate as well. I have no hesitation to recommend straightforward publication.
Ans: We really appreciate your encouragement. We will keep working on the more detailed related research.
I have only few suggestions. The authors can improve the Introduction section with recent paper published in literature regarding the risk of VTE, as following:
- https://doi.org/10.3390/medicina58101502
- https://doi.org/10.3390/jcm9123970
- https://doi.org/10.3390/diagnostics12112757
- https://doi.org/10.3390/jcm9051257
Ans: Thank you for your suggestion. We have added the literature you suggested in Introduction as below:” …… However, VTE incidence has been reported to vary considerably in different populations and countries [7,8]. WoÅ‚kowski et al in 2020 claimed that the sum score of the patient and surgical procedure characteristics was important in the evaluation of the risk of VTE [9]. In this period of COVID-19 pandemic, MureÈ™an et al in 2022 concluded that nutrition status, peripheral arterial disease, malignancy and tobacco may predict the of risk and mortality of PE and DVT among COVID-19 patients [10]. Although the pharmacological prophylaxis….. Knowing the risk factors related to the perioperative VTE is very important for orthopedic surgeons, especially when they plan to perform the elective operations for the patients, such as total joint arthroplasty or thoracolumbar spine surgeries, etc. Kawai et al in 2020 and Melinte et al in 2022 revealed that older age, malignancy, cardiovascular disease, obesity, tobacco, steroid and low preoperative function score may be the important risk factors for the incidence of DVT during total hip or knee arthroplasty [13,14]. Identifying populations with a higher risk of postoperative VTE…...”
